# Prenatal Diagnosis of Right-Sided Congenital Ventricular Diverticulum (CVD) Assisted by Spatiotemporal Image Correlation (STIC) Acquisition and the Speckle-Tracking Technique to Assess Fetal Cardiac Function: A Case Report

**DOI:** 10.3390/diagnostics12102438

**Published:** 2022-10-08

**Authors:** Ji Hye Koh, Hyo Jeong Lee, Jinyoung Choi, Jun Woo Ahn

**Affiliations:** 1Department of Obstetrics and Gynecology, University of Ulsan College of Medicine, Asan Medical Center, Seoul 05505, Korea; 2Department of Obstetrics and Gynecology, University of Ulsan College of Medicine, Ulsan University Hospital, Ulsan 44033, Korea

**Keywords:** congenital ventricular diverticulum, prenatal diagnosis, spatiotemporal image correlation, STIC, speckle tracking, *Fetal*HQ

## Abstract

Congenital ventricular diverticulum (CVD) is a cardiac malformation defined as an outpouching lesion of a ventricle with normal contractility and thickness, and the advancement of prenatal sonography has led to its prenatal diagnosis. In the prenatal period, CVD is reported in association with pericardial effusion, arrhythmia, and fetal hydrops or as an isolated condition. With the development of prenatal echocardiography, CVD can be diagnosed from the early stage of pregnancy, and prenatal procedures, such as pericardiocentesis, are also possible. Spatiotemporal image correlation (STIC) acquisition, a novel approach for the clinical evaluation of fetal hearts, provides easy-to-use techniques for acquiring data from fetal hearts and helping visualization with two-dimensional and three-dimensional (3D) cine sequences. Furthermore, the speckle-tracking technique enables a more comprehensive evaluation of the shape, continuity, and function of the fetal heart. These recent techniques have never been used for CVD diagnosis and evaluation. Here, we present a case of right-sided CVD, which is the first in which STIC acquisition and cardiac function assessment with the speckle-tracking technique were used to assist in the diagnosis and evaluation.

## 1. Introduction

Congenital ventricular diverticulum (CVD) is a cardiac malformation defined as an outpouching lesion of a ventricle with normal contractility and thickness, and the advancement of prenatal sonography has led to its prenatal diagnosis [1,2]. In the prenatal period, CVD is reported in association with pericardial effusion, arrhythmia, and fetal hydrops or as an isolated condition [2,3,4,5]. When a ventricle shows outpouching, it is necessary to differentiate CVD from congenital ventricular aneurysm (CVA). Histologically, a CVD lesion contains the myocardium in its wall, while a CVA lesion lacks the myocardium within its wall but contains fibrous tissue [2]. Ultrasonography is the only prenatal method to distinguish between the two conditions, and sonographic findings of CVD typically include an outpouching ventricle with a saccular shape, a narrow base and normal contractility [2]. Sonographic techniques such as B-mode and M-mode ultrasonography have been used to identify CVD [6]. Spatiotemporal image correlation (STIC) acquisition, a novel approach for the clinical evaluation of fetal hearts, has never been used for CVD diagnosis and evaluation. STIC is an easy-to-use technique for acquiring data from fetal hearts and helping visualization with two-dimensional and three-dimensional (3D) cine sequences [7]. Fetal heart images acquired by STIC volume are known to be superior for evaluating ventral wall motion and valve excursion [8]. Furthermore, the speckle-tracking technique enables a more comprehensive evaluation of the shape, contractibility, and function of the fetal heart, especially when the GE Automatic Fetal Heart Assessment Tool, *Fetal*HQ (GE Healthcare Korea, Seoul, Korea), is available [9]. Here, we present a case of right-sided CVD, which is the first case in which STIC acquisition and cardiac assessment with speckle-tracking echocardiography were used to assist in the diagnosis.

## 2. Case

A 43-year-old pregnant woman with dichorionic diamniotic (DCDA) twins was referred due to abnormal fetal ultrasonographic findings for one of the twins. At a gestational age (GA) 26^+2^ weeks, fetal echocardiography of the upper fetus showed an enlarged and globular right ventricle (RV) with midsystolic tricuspid regurgitation (TR) when compared with the findings for the lower fetus (Figure 1A,B). Except for the ventricular differences and TR, the cardiac structures of the upper fetus were normal, and there were no associated abnormalities. In early pregnancy, the mother underwent genetic amniocentesis because of her advanced age, and the karyotypes of both fetuses were normal. Additionally, the detailed sonographic findings of the lower fetus were normal. At a GA of 30^+5^ weeks, the globular shape of the RV remained unchanged, but holosystolic TR was observed (Figure 1C,D). Since there was a risk of fetal hydrops or demise due to the progression of TR, Doppler testing, including the ductus venosus, was performed, and the results were normal (Figure 1E).

At a GA of 35^+2^ weeks, the globular RV of the upper fetus persisted (Figure 2A). Because a rare cardiac chamber disease was suspected, a decision to consult a pediatric cardiologist was made. After consultation, right-sided ventricular diverticulum was diagnosed. Although the outpouching portion of the ventricular wall was observed to be wider than previously reported, ventricular diverticulum was diagnosed because of the synchronous contraction of the wall and the intact thickness of the ventricle. The evaluation of ventricular thickness and synchronous contraction was assisted by the analysis of the acquired STIC volume. Cardiac images with STIC more markedly indicated these muscle layers than B-mode images, and synchronous coordination was more clearly observed in the video of the cardiac cycle with STIC (Figure 2B,C, Appendix A).

Since the new ultrasound device, a Voluson E10 with an eM6C G2 metrix probe (GE Healthcare Korea, Seoul, Korea), had been introduced to our hospital, quantitative assessment of the fetal heart using *Fetal*HQ was performed. First, in the four-chamber view (4CV) global size and shape, the global sphericity index (GSI) was measured. B-mode images were used for the previous two scans, and speckle-tracking analysis was used for the latter two scans (Figure 3 and Table 1).

In the fetus with CVD, the GSI was continuously measured below the fifth percentile [10]. Second, using *Fetal*HQ, the size and shape of each ventricle were evaluated in 24-segment ventricular analysis and 24-segment sphericity index (SI) analysis, respectively. Twenty-four-segment ventricular analysis involves the evaluation of transverse widths by dividing the ventricle into segments at end diastole. At a GA of 35^+2^ weeks, the analysis of the fetus with CVD showed that the RV was wider than the reference value and was particularly prominent in the base (1–8) and mid (9–16) regions, while the LV was well in line with the median z score (Figure 4B upper). These findings were commonly observed at 33^+5^ weeks GA (Appendix A). The 24-segment sphericity index (SI) is used as an indicator of diastolic shape and is calculated by dividing the ventricle into segments. At a GA of 35^+2^ weeks, the fetus with CVD had an all but one of SI value in the base segment (1–8) that was below the tenth percentile, while the value of the SI in the LV was in the normal range (Figure 4A,B lower graph). This result indicated that the base region of the RV had an abnormal globular shape. These findings had also been continuously observed before (Appendix A). On the other hand, the healthy lower fetus had a 24-segment SI in both ventricles that was greater than the 10th percentile at the same GA (Figure 4C).

Cardiac function parameters were computed by *Fetal*HQ with STIC volume acquisition, and the results are shown in Table 2 and Figure 5. The ejection fraction (EF) of the fetus with CVD decreased from 46% at a GA 33^+5^ weeks to 42% at a GA 35^+2^ weeks, whereas the EF of the healthy fetus was 46% at a GA of 35^+2^ weeks. The global longitudinal strain (GLS) in the RV of the fetus with CVD decreased from −6.43% to −4.97%, while the LV GLS increased from −2.96% to −5.53%. Regardless of the ventricle, the GLS of the fetus with CVD was lower than that of the healthy fetus (Table 2).

At a GA of 36^+6^ weeks, an emergency cesarean section was performed due to preterm labor. The weight of the first baby (lower healthy fetus) was 2220 g, and the weight of the second baby (upper fetus with CVD) was 2550 g. The fetuses were admitted to the neonatal intensive care unit due to prematurity. The second baby did well without any abnormal symptoms, and an echocardiography was performed on the first day of life. In the echocardiography, a large RV diverticulum at the anterior free wall below the tricuspid valve annulus was observed, and the paradoxical motion of interventricular septum with a preserved wall thickness and good LV function (LV ejection fraction 62%) was noted (Appendix A). A small atrial septum defect (4 mm) and TR were also observed. Since there was the possibility of heart surgery, heart computed tomography (CT) was performed. The heart CT indicated a heart axial plane similar to the antenatal 4CV (Appendix A). During the six months leading up to the last follow-up, the baby did not show any problems with growth, showed no symptoms, and did not undergo surgery or intervention. The last echocardiography was performed at the age of 6 months, and the echocardiography findings were similar to those observed previously.

## 3. Discussion

Ventricular diverticulum is a rare cardiac malformation, and one report estimated its approximate incidence to be 0.013% [6]. Although one case of RV diverticulum was reported at a GA of 12 weeks, most cases (75%) were diagnosed after the first trimester [2]. According to a recent study, among the 40 reported cases, only two cases of CVD occurred in multiple pregnancies, which were monochorionic diamniotic twins [2,11]. Therefore, this case is the first reported CVD case in a DCDA twin. Notably, to the best of our knowledge, our report is the first in which STIC technology was used in the diagnosis and the first case in which ventricle geometry and cardiac functional parameters were evaluated using speckle tracking.

If a protrusion of the ventricle wall is observed on fetal echocardiography, ventricular aneurysm and CVD should be differentiated. The echocardiographic features used in the prenatal diagnosis of CVD include a narrow size of the communication with the ventricle, intact ventricular thickness, and normal contractility. Regarding communication size with the ventricle in CVD, there were cases in which the size was reported to be large, or the size measurement was inadequate because of location, so there is a limit of communication size as a differentiation point [6,12]. The evaluation of the integrity of the wall, which has a similar thickness and layering as the rest of the ventricle in patients with CVD, is made by B-mode ultrasonography [12]. Last, synchronous movement of the wall, which is observed in patients with CVD, is confirmed through M-mode ultrasonography [6].

STIC in fetal cardiac imaging, the most recent technology, reconstructs the entire fetal heart within a cardiac cycle and can show 3D ultrasound images in various imaging modes [13]. In particular, images using STIC are excellent at evaluating ventral wall motion and valve excursion [8]. Comparing the B-mode 2D image and the image acquired by STIC in this fetus, it is clear that all muscle layers are included in the more protruding ventricle in the STIC image (Figure 2). Moreover, depending on the fetal position, it may be difficult to evaluate the overall thickness of the ventricular wall, especially as this is more challenging in multiple pregnancies. STIC enables spatial and temporal postprocessing using a fully reconstructed cardiac cycle in volume information. Given this ability to rotate the fetal heart along the *z*-axis in STIC volume analysis, the examiner can position the heart to allow examination in various axes. In this case, the ventricular diverticulum was evaluated from various planes of heart (Appendix A). In particular, the 4D image of the CVD lesion clearly identified synchronous ventricle movement (Appendix A). In addition, the video clip made it possible to explain the heart abnormalities of the fetus to the parent, who had little medical knowledge, and it was helpful for prenatal counseling.

Pericardial effusion (PE) is one of the most common symptoms accompanying CVD and is observed in 60% of patients according to recent reports [2]. The presence of CVD leads to concerns about heart failure because the effusions compress the heart. Thus, pericardiocentesis was performed to decompress the fetal thorax [2,14,15]. However, an evaluation of heart function in a patient with CVD was performed in only one case report [15]. In this study, cardiac function parameters of CVD were measured using *Fetal*HQ with the STIC acquisition in order to evaluate heart function. The functional parameters, listed in Table 1 and Table 2, indicate the global ventricular contractility. The ejection fraction is indicative of the systolic function of the LV, and fetuses with CVD have been observed to have lower EFs than healthy fetuses. According to a study evaluating fetal EF using speckle tracking, both fetuses had EFs that were lower than the fifth percentile using the estimated fetal weight as the independent variable [16]. Global longitudinal strain (GLS) is known to reflect myocardial deformation and can be used to assess cardiac function. According to research on the fetal GLS reference range, GSL in the left ventricle decreases with increasing GA, while there is no change in the GSL in the RV with GA [17]. Therefore, the GSL in the LV of the fetus with CVD normally increased with increasing GA, from −2.96% to −5.53%. On the other hand, the GSL in the RV should remain unchanged, but since it decreased, there was a possibility of RV myocardial deformation. In any week of pregnancy, the global longitudinal strain in the LV in fetuses with cardiac abnormalities was less than 5% according to reference GLS values [17]. The lower healthy twin showed a global longitudinal strain value corresponding to the 10th percentile at a GA of 35 weeks. Since there is no previous report on the cardiac parameters of fetuses with ventricular diverticula, the interpretation of the values is very limited. In the future, as more research is conducted, it will be helpful to predict the prognosis, such as progression to hydrops, by confirming changes in the heart function of patients with CVD.

It is known that in various disease states, the fetal heart not only increases the thickness of the ventricles and septum but also adapts by changing the size of the atria and ventricles [10]. The global spherical index (GSI) has been identified as an indicator of this change in cardiac shape [18]. The GSI is calculated by measuring the basal-apical length (BAL) and transverse length (TL) in a four-chamber view during the end diastole period. The GSI is calculated by dividing the BAL by the TL. In this fetus, in the first two ultrasound scans, the author calculated the GSI from the 2D image, and the *Fetal*HQ with STIC volume was used in the latter two ultrasound scans. STIC imaging was used because the operators can easily display diagnostic planes with STIC acquisition [8]. When the GSI is below the fifth percentile, it can be assumed that the cardiac shape is abnormal and globular [10]. In this fetus, all GSIs were below the 5th percentile, especially in comparison to the other twin. The other measurement, 24-segment ventricular analysis, analyzes the transverse widths of the ventricle by dividing it into 24 segments. The ventricular width changes are evaluated at the base, mid, and apex regions, and the values can be compared with the z score. In this fetus, a 24-segment ventricular analysis showed that the RV diverticulum lesion and the base and mid regions deviated more from the normal range than the LV and the RV apex region. Notably, the size of the RV of the upper fetus was measured to be very large compared to the size of the RV of the lower fetus, who had a normal heart. However, there have been no reports of ventricle shape in ventricular diverticulum cases, especially with 24-segment ventricular analysis and GSI. Therefore, there are limitations in interpreting the measured values at this time.

In conclusion, STIC volume acquisition and *fetal*HQ evaluated the shape and function of the diverticulum heart in this case, and the parameter summary is as follows. First, a decreased global longitudinal strain of RV implied ventricle dysfunction, which originated from myocardial deformation. Second, the global spherical index showed an abnormal global heart shape. Lastly, the 24-segment ventricular analysis of RV showed the widely expanded ventricular wall numerically. This report is the first in which the STIC technique and f *fetal*HQ using the speckle-tracking technique were used in diagnosing and evaluating CVD, and if many studies are conducted in the future with those techniques, it will greatly contribute to CVD diagnosis and prognosis evaluation.

## Figures and Tables

**Figure 1 diagnostics-12-02438-f001:**
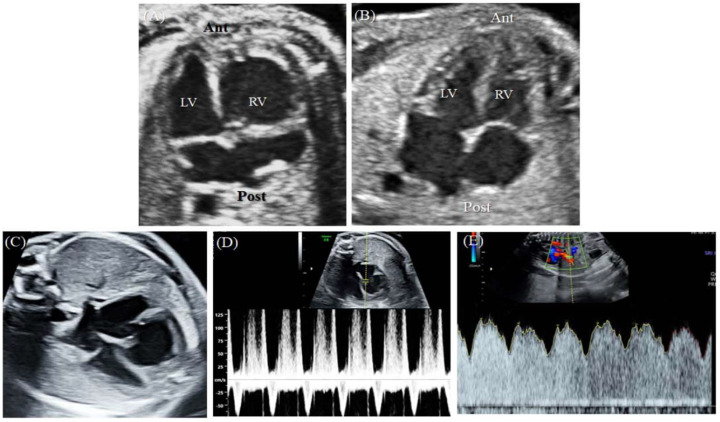
This is a figure. Schemes follow the same formatting.(**A**) At 26^+2^ weeks GA, the upper fetus showed globular RV enlargement compared with (**B**) the lower fetus, who had a normal heart. (**C**) At 30^+5^ weeks GA, the globular shape of the RV remained unchanged, but (**D**) holosystolic TR was observed. (**E**) Doppler evaluation of the ductus venosus showed normal findings. Ant, anterior; Lt, left; LV, left ventricle; Post, posterior; Rt, right; RV, right ventricle.

**Figure 2 diagnostics-12-02438-f002:**
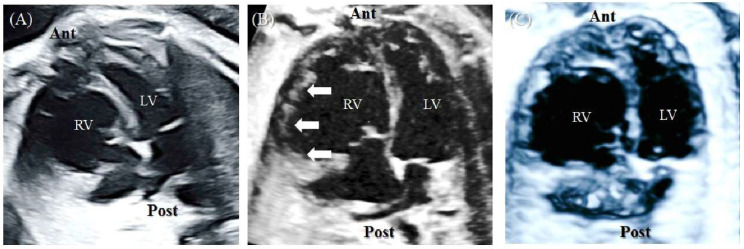
At 35^+2^ weeks GA, fetal echocardiography was performed. (**A**) B-mode image of four-chamber view. The globular RV persisted. (**B**) The cardiac image with STIC technology. The outpouching ventricular wall (arrowhead) was observed and had a thickness and layering similar to those of the rest of the ventricle. (**C**) Video still image of the 4D video, which was acquired by surface-rendering mode of the STIC volume in grayscale. Ant, anterior; Lt, left; LV, left ventricle; Post, posterior; Rt, right; RV, right ventricle; STIC, spatiotemporal image correlation.

**Figure 3 diagnostics-12-02438-f003:**
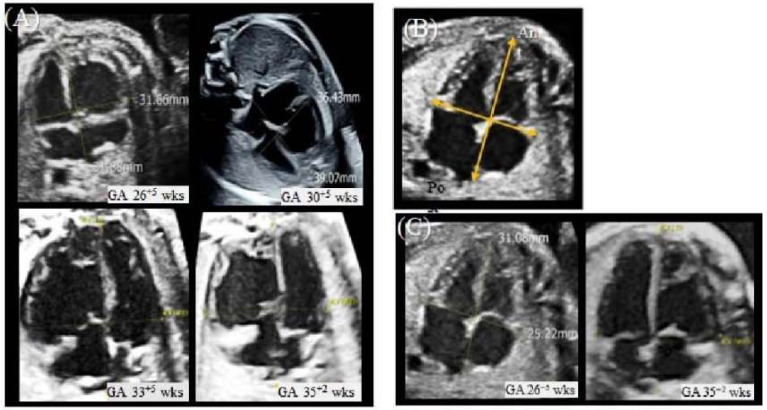
Measurement of the GSI.(**A**) The measurement of the GSI in the fetus with CVD at each gestational age. The first two scans (GA26^+5^ weeks, GA 30^+5^ weeks) were performed with B-mode imaging. The latter two scans (GA33^+5^ weeks, GA35^+5^ weeks) were performed with STIC imaging with *Fetal*HQ. (**B**) The measurement of the GSI in the first two scans (GA26^+5^ weeks, GA 30^+5^ weeks). In the 4CV, the basal-apical length (BAL) was measured at end-ventricular diastole from the epicardial border of the posterior atrial wall to the epicardial border of the apical ventricular wall. Transverse length (TL) at the widest part in the 4CV from the epicardial borders of the ventricular walls. The GSI was calculated formally with BAL/TL, and it was compared with the z score referenced in DeVore 2017. (**C**) The measurement of the GSI in the healthy fetus at each gestational age. The first scan (GA26^+5^ weeks) was performed with B-mode imaging. The latter scan (GA33^+5^ weeks, GA35^+5^ weeks) was performed with STIC imaging with *Fetal*HQ. EFW, estimated fetal weight; GA, gestational age; GSI, global sphericity index; wks, weeks.

**Figure 4 diagnostics-12-02438-f004:**
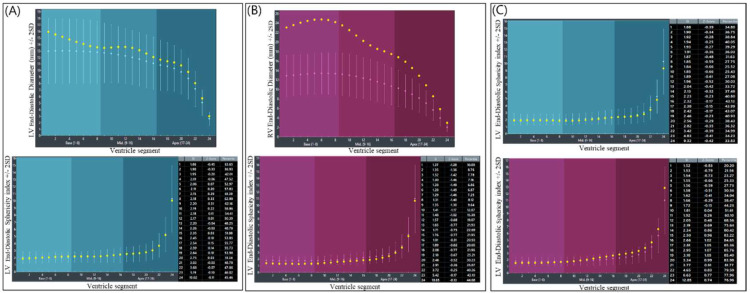
Twenty-four-segment ventricular analysis and the 24-segment sphericity index at a GA 35^+2^. (**A**) LV geometry of the fetus with CVD. The upper graph shows the results of the 24-segment ventricular analysis. The lower graph shows the results of the 24-segment sphericity index analysis. (**B**) RV geometry of the fetus with CVD. The upper graph shows the results of the 24-segment ventricular analysis. The lower graph shows the results of the 24-segment sphericity index analysis. The RV was wider than the reference value and was particularly prominent at the base (1–8) and mid (9–16) regions, the widths of which were out of range. (**C**) Twenty-four-segment sphericity indices of the healthy fetus. All measurements were above the 10th percentile. LV, left ventricle; Rt, right; RV, right ventricle; SD; standard deviation.

**Figure 5 diagnostics-12-02438-f005:**
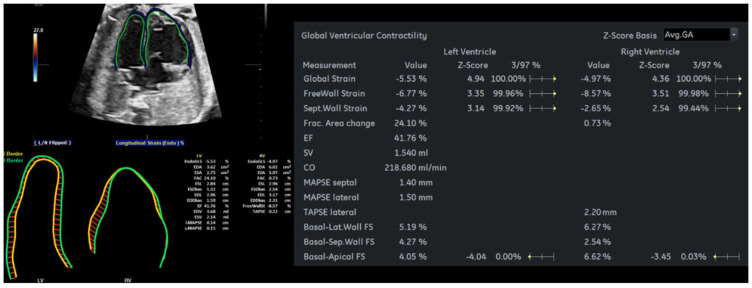
*Fetal*HQ results of the fetus with CVD using the speckle-tracking technique at a GA 35^+2^ wks. CO, cardiac output; EdoGLS, endocardium global strain; EDA, end-diastolic area; EDS, end-systolic area; EF, ejection fraction; ESL, end-systolic length; ESDbas, end-systolic displacement of basal; FAC, fractional area change; GA, gestational age; Lat, lateral; LV, left ventricle; MAPSE, mitral annular plane systolic excursion; RV, right ventricle; Sep, septal; SV, stroke volume; TAPSE, tricuspid annular plane systolic excursion; wks; weeks.

**Table 1 diagnostics-12-02438-t001:** (A). CVD fetus (upper fetus). (B). Normal fetus (lower fetus).

**(A) CVD Fetus (Upper Fetus)**					
**GA**	**EFW(g)**	**Basal-Apical Length (mm)**	**Transverse Length (mm)**	**GSI**	**Percentile (%)**
26.5	866	31.88	31.66	1.007	0.885
30.5	1562	39.07	36.43	1.072	4.604
33.5	2013	43.17	44.42	0.97	0.307
35.2	2446	46.89	46.89	1	0.724
**(B). Normal Fetus (Lower Fetus)**					
**GA**	**EFW(g)**	**Basal-Apical Length (mm)**	**Transverse Length (mm)**	**GSI**	**Percentile (%)**
26.5	756	31.08	6.7	1.232	49.73
30.5					
33.5					
35.2	1936	40.15	35.46	1.13	14.524

**Table 2 diagnostics-12-02438-t002:** (A) Cardiac functional parameters of the fetus with CVD obtained by *Fetal*HQ; (B) Cardiac functional parameters of the healthy fetus obtained by CO, cardiac output. EdoGLS, endocardium global strain; EDA, end-diastolic area; EDS, end-systolic area; EF, ejection fraction; ESL, end-systolic length; ESDbas, end-systolic displacement of basal; FAC, fractional area change; GA, gestational age; Lat, lateral; LV, left ventricle; MAPSE, mitral annular plane systolic excursion; RV, right ventricle; Sep, septal; SV, stroke volume; TAPSE, tricuspid annular plane systolic excursion; wks; weeks.

**(A)**	**GA33^+5^ wks**		**GA35^+2^ wks**	
	**LV**	**RV**	**LV**	**RV**
Global Strain (%)	−2.96	−5.53	−6.43	−4.97
Freewall Strain (%)	−1.15	−6.77	−11.23	−8.57
FAC (%)	26.51	24.1	2.7	0.73
EF (%)	45.83	41.76		
**(B)**	**GA35^+2^ wks**			
	**LV**		**RV**	
Global Strain (%)	−15.78		−7.72	
Freewall Strain (%)	−15.96		−4.98	
FAC (%)	31.64		9.49	
EF (%)	46.28			

## Data Availability

Not applicable.

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
