# Peer review of "Prenatal Diagnosis of Right-Sided Congenital Ventricular Diverticulum (CVD) Assisted by Spatiotemporal Image Correlation (STIC) Acquisition and the Speckle-Tracking Technique to Assess Fetal Cardiac Function: A Case Report"

_diagnostics, 2022, doi:10.3390/diagnostics12102438_

Round 1
Reviewer 1 Report
In this study the authors report a case of prenatal diagnosis of RV diverticulum. They reported anatomic and functional prenatal evaluation. The case interesting. Some comments have to be addressed.
- A fetal or neonatal TTE video could be useful to illustrate and support the diagnosis of the RV diverticulum
- What was the indication for CT in this neonate
- The authors reported “while a CAV lesion lacks the myoca dium within its wall but contains fibrous tissue, please correct to CVA”
- was the PFO shunt normal during e fetal life?
Author Response
Thank you very much for your review of our manuscript. The comments were constructive and helped us to revise and improve our document. The following is an itemized account of the changes made in response to your comments.
Point#1:
A fetal or neonatal TTE video could be useful to illustrate and support the diagnosis of the RV diverticulum
Response 1:
Thank you very much for your comment. We add neonatal TTE video with supplementary video S3. Unfortunately, we had very short video of fetal cardiac image, which is very similar to video S1.
Point#2
What was the indication for CT in this neonate
Response 2:
It was a decision of the pediatric cardiologist. Since there is a possibility of heart surgery, it was performed to prepare for this. We add followed sentence to clarify this in line number 173.
“ Since there was possibility of heart surgery, heart computed tomography (CT) was performed.”
Point#3
The authors reported “while a CAV lesion lacks the myoca dium within its wall but contains fibrous tissue, please correct to CVA
Response 3:
Thank you very much for your comment. We just fixed it in the line number 35.
Point#4
- was the PFO shunt normal during e fetal life?
Response4:
Yes. Although TR was accompanied, there was no abnormality in shunting flow through PFO. Also, you could find the membrane of foramen ovale flap in Figure 2 (B).
Reviewer 2 Report
There is no conclusion to your article. Why ?? You need to summarize all the clinical data and sum it in a conclusion. Make it look excellent. Your conclusion bring the beauty of your writing style. Everything else looks very good.
Author Response
Thank you very much for your review of our manuscript. The comment was very constructive and helped us to revise and improve our document. The following is an itemized account of the changes made in response to your comments.
Point#1:
There is no conclusion to your article. Why ?? You need to summarize all the clinical data and sum it in a conclusion. Make it look excellent. Your conclusion bring the beauty of your writing style. Everything else looks very goo
Response:
I changed the last paragraph to followed sentences :) and the changed manuscrip is attached.
In conclusion, STIC volume acquisition and fetalHQ evaluated the shape and function of the diverticulum heart in this case, and the parameter summary is as follows. First, a decreased global longitudinal strain of RV implied ventricle dysfunction, which originated from myocardial deformation. Second, the global spherical index showed an abnormal global heart shape. Lastly, the 24-segment ventricular analysis of RV showed the widely expanded ventricular wall numerically. This report is the first in which the STIC technique and f fetalHQ using the speckle-tracking technique were used in diagnosing and evaluating CVD, and if many studies are conducted in the future with those technique, it will greatly contribute to CVD diagnosis and prognosis evaluation.

Reviewer 3 Report
Authors have studied prenatal diagnosis of right-sided congenital ventricular diverticulum based on spatiotemporal image correlation. Authors might draw reader’s attention and new insights are highlighted. Moreover, , 10.1007/s11071-021-06517-w; 10.1007/s11071-021-06262-0 . Conclusion part can be improved by making precise. I would suggest for minor revision before recommendation.
Author Response
Thank you very much for your review of our manuscript. The comment was very constructive and helped us to revise and improve our document. The following is an itemized account of the changes made in response to your comments.
Point#1:
Authors have studied prenatal diagnosis of right-sided congenital ventricular diverticulum based on spatiotemporal image correlation. Authors might draw reader’s attention and new insights are highlighted. Moreover, , 10.1007/s11071-021-06517-w; 10.1007/s11071-021-06262-0 . Conclusion part can be improved by making precise. I would suggest for minor revision before recommendation.
Response:
I changed the last paragraph to followed sentences, and the changed manuscrip is attached.
In conclusion, STIC volume acquisition and fetalHQ evaluated the shape and function of the diverticulum heart in this case, and the parameter summary is as follows. First, a decreased global longitudinal strain of RV implied ventricle dysfunction, which originated from myocardial deformation. Second, the global spherical index showed an abnormal global heart shape. Lastly, the 24-segment ventricular analysis of RV showed the widely expanded ventricular wall numerically. This report is the first in which the STIC technique and f fetalHQ using the speckle-tracking technique were used in diagnosing and evaluating CVD, and if many studies are conducted in the future with those technique, it will greatly contribute to CVD diagnosis and prognosis evaluation.

Reviewer 4 Report
Koh et al present a case report regarding use of spatiotemporal image correlation to improve fetal imaging. Methodology appears valid and is likely reproducible. Details are provided on the application of the technique towards a challenging prenatal diagnosis.
Author Response
Thank you very much for your review of our manuscript. According to the other reviewer's comment, we add a conclusion part. The changed manuscript is attached.

Round 2
Reviewer 1 Report
The authors answered correctly to the comments.